# Plasma Levels of snoRNAs are Associated with Platelet Activation in Patients with Peripheral Artery Disease

**DOI:** 10.3390/ijms20235975

**Published:** 2019-11-27

**Authors:** Anne Yaël Nossent, Neda Ektefaie, Johann Wojta, Beate Eichelberger, Christoph Kopp, Simon Panzer, Thomas Gremmel

**Affiliations:** 1Department for Laboratory Medicine, Medical University of Vienna, 1090 Vienna, Austria; a.y.nossent@lumc.nl; 2Department of Internal Medicine II, Medical University of Vienna, 1090 Vienna, Austriajohann.wojta@meduniwien.ac.at (J.W.); christoph.kopp@meduniwien.ac.at (C.K.); 3Department of Surgery, Leiden University Medical Center, 2300 RC Leiden, The Netherlands; 4Einthoven Laboratory for Experimental Vascular Medicine, Leiden University Medical Center, 2300 RC Leiden, The Netherlands; 5Core Facilities, Medical University of Vienna, 1090 Vienna, Austria; 6Ludwig Boltzmann Cluster for Cardiovascular Research, 1090 Vienna, Austria; 7Department of Blood Group Serology and Transfusion Medicine, Medical University of Vienna, 1090 Vienna, Austria; 8Department of Internal Medicine, Cardiology and Nephrology, Landesklinikum Wiener Neustadt, 2700 Wiener Neustadt, Austria

**Keywords:** peripheral artery disease, restenosis, angioplasty, snoRNA, platelets

## Abstract

In addition to supervised walking therapy, antithrombotic therapy and the management of risk factors, the treatment of peripheral artery disease (PAD) is limited to endovascular and surgical interventions, i.e., angioplasty with stent implantation and bypass surgery, respectively. Both are associated with a high restenosis rate. Furthermore, patients with PAD often suffer atherothrombotic events like myocardial infarction, transient ischemic attacks or stroke. Small ribonucleic acids (RNAs) have proven reliable biomarkers because of their remarkable stability. Small nucleolar RNAs (snoRNAs) guide modifications to small nuclear RNAs and ribosomal RNAs, enabling protein synthesis. In the current study, we measured four snoRNAs in 104 consecutive PAD patients who underwent elective infrainguinal angioplasty with stent implantation. We selected snoRNAs that showed significant overexpression in the plasma of end-stage PAD patients in a previous study. All four snoRNAs are transcribed from the 14q32 locus, which is strongly linked to human cardiovascular disease, including PAD and restenosis. We showed that the four selected 14q32 snoRNAs were abundantly expressed in the plasma of PAD patients. The plasma levels of these snoRNAs were not directly associated with target vessel restenosis, however, levels of SNORD113.2 and SNORD114.1 were strongly linked to platelet activation, which is an important determinant of long-term outcome, in PAD, and in cardiovascular disease in general.

## 1. Introduction

Peripheral artery disease (PAD) is caused by progressive atherosclerosis and is most frequently seen in the arteries of the lower extremities [1]. In earlier stages of the disease, stenoses of the femoral and/or popliteal arteries often leads to insufficient blood supply during exercise, causing intermittent claudication. As the disease progresses and stenoses of the arteries increase, many patients experience pain at rest, referred to as critical limb ischemia. Patients with critical limb ischemia are at risk of developing non-healing wounds, ulcers and tissue necrosis often requiring amputation of the affected tissue [2].

In addition to supervised walking therapy, antithrombotic therapy and the management of cardiovascular risk factors, including smoking, hypertension, obesity and hypercholesterolemia, PAD treatment is limited to endovascular and surgical interventions, i.e., angioplasty with stent implantation and bypass surgery, respectively. Unfortunately, both are associated with a high restenosis/reocclusion rate [3]. Furthermore, patients with PAD often suffer atherothrombotic events like myocardial infarction (MI), transient ischemic attacks (TIA) or stroke [2,4,5].

Accordingly, it is important to closely monitor PAD patients and to thoroughly understand risk factors and biomarkers for disease progression, postinterventional restenosis and atherothrombotic outcomes. In recent years, several studies, including work from our own group, have investigated the potential of microRNAs as biomarkers for PAD progression and restenosis [6,7]. MicroRNAs have proven to be remarkably stable in the circulation in vivo, but also in blood, plasma and serum samples ex vivo, even after long-term storage and repeated freeze-thaw cycles [8]. However, microRNAs are not the only small ribonucleic acids (RNAs) that display such favourable properties for biomarkers. We recently showed that a set of small nucleolar RNAs (snoRNAs) had potential as biomarkers for muscle damage and repair in elite athletes and end stage PAD patients [9], as well as for the recovery of patients with ST-elevation myocardial infarction (STEMI) that underwent percutaneous coronary intervention (PCI) [10]. SnoRNAs are also small RNA molecules that are between 60 and 300 nucleotides in length and play an important role in cellular machinery. They guide crucial stabilizing modifications to small nuclear RNAs (snRNAs) and ribosomal RNAs (rRNAs), allowing efficient pre-mRNA splicing and protein synthesis, respectively [11].

In the current study, we measured four snoRNAs in 104 consecutive PAD patients who underwent elective infrainguinal angioplasty with stent implantation. We selected snoRNAs that showed significant overexpression in the plasma of end-stage PAD patients compared to healthy individuals in a previous study [9]. All four snoRNAs are transcribed from the 14q32 locus, a noncoding RNA gene locus that is strongly linked to human cardiovascular disease, including PAD and restenosis [10,12,13,14,15,16]. We aimed to evaluate the associations between snoRNA plasma levels with restenosis, MI and TIA/stroke, but also with classical risk factors for PAD development and progression, including smoking and platelet activation.

## 2. Results

### 2.1. SnoRNA Plasma Levels

We previously published that the expression of SNORD112, SNORD113.2 and SNORD113.6 was significantly higher in PAD patients with critical ischemia than in healthy controls [9]. While plasma levels were generally around the detection limit in controls, all four snoRNAs were abundantly expressed in PAD patients with critical ischemia. In the current study, snoRNAs were readily detectable in the plasma of 101, 99, 95 and 101 of the 104 patients for SNORD112, SNORD113.2, SNORD113.6 and SNORD114.1, respectively. SnoRNA plasma levels were excessively high in 1, 6, 4 and 4 patients for SNORD112, SNORD113.2, SNORD113.6 and SNORD114.1, respectively. These patients were excluded from further analyses (Figure 1).

### 2.2. Clinical Endpoints

Out of 104 patients, 41 (39.4%) reached the primary clinical endpoint within the 24-month follow-up period; 36 patients (34.6%) developed target vessel restenosis, 3 patients (2.9%) had a TIA/stroke and 2 patients (1.9%) had a myocardial infarction. Taking these endpoints together, there were no associations between snoRNA levels and the primary composite endpoint. We then looked at the different clinical endpoints separately. There were no associations between snoRNA levels and the individual endpoints, target vessel restenosis/reocclusion, MI or TIA/stroke either. Kaplan–Meier analyses also failed to show any association between low or high snoRNA levels and the time-to-endpoint.

### 2.3. Classical Risk Factors

We compared snoRNA levels between PAD patients with or without hypertension, diabetes, and hyperlipidemia using student’s t-tests. No associations were observed. We then performed linear regression on the analyses to assess potential correlations between snoRNA levels and age, body mass index (BMI), total cholesterol, triglycerides, LDL, HDL, lipoprotein (a) (Lpa), CRP levels and total leukocytes. SNORD112 showed a significant inverse correlation with Lpa (B = −0.228, *p* = 0.025). We then performed a *t*-test to compare Lpa levels between PAD patients with low and high SNORD112 levels (Lpa = 72.47 vs. 50.80, respectively; *p* = 0.038). Furthermore, SNORD113.2 showed a significant correlation with triglyceride levels (B = 0.219, *p* = 0.034). We then compared triglyceride levels between patients with low vs. high SNORD113.2 levels (Triglycerides= 155.49 vs. 213.56, respectively; *p* = 0.27).

Furthermore SNORD113.6 showed a small but significant correlation with total leukocytes (B = 0.012, *p* = 0.026), whereas SNORD113.2 showed a trend towards an inverse correlation with total leukocytes (B = −0.746, *p* = 0.098). We then compared total leukocytes between patients with low vs. high snoRNA levels. Differences in leukocytes between patients with low vs. high SNORD113.6 were too small to remain significant and as for linear regression, there were no differences in leukocytes between patients with low vs. high SNORD112. However, a clear trend remained for low vs. high SNORD113.2 and a similar trend was observed for SNORD114.1 (Leukocytes= 9.188 vs. 8.418 for low vs. high SNORD113.2, respectively, *p* = 0.10; Leukocytes= 9.119 vs. 8.394 for low vs. high SNORD114.1, respectively, *p* = 0.057).

### 2.4. SnoRNA Levels and Smoking

Smoking is one of the most prominent risk factors for PAD. However, it is also a somewhat controversial risk factor, as smoking has been shown to reduce the risk of post-interventional restenosis [3]. In our population, there was no significant association between smoking and target vessel restenosis. However, although not significant, there were fewer active smokers in the group of patients that developed restenosis than in the group that did not (37% vs. 47%, respectively; *p* = 0.221). Furthermore, there was a trend towards a longer time-to-endpoint in the group of active smokers (*p* = 0.076). Smokers also showed a trend towards reduced platelet activation, as the platelet surface expression of activated GPIIb/IIIa appeared decreased in active smokers compared to non-smokers, both in vivo and after stimulation with ADP (2.74 vs. 3.03 for activated GPIIb/IIIa in vivo in smokers vs. non-smokers, respectively; *p*= 0.076; 10.71 vs. 13.28 for activated GPIIb/IIIa in response to ADP in smokers vs. non-smokers, respectively; *p* = 0.085).

We compared snoRNA levels between smokers and non-smokers and found that active smokers had significantly lower levels of SNORD114.1 than non-smokers (5.48 vs. 8.68, respectively; *p* = 0.031). When comparing patients with low vs. high snoRNA levels, we observed that there were less smokers in the group of patients with high SNORD113.2 levels, compared to Low SNORD113.2 levels (34% vs. 55%, respectively; *p* = 0.044) and in the group of patients with high SNORD114.1 levels compared to low SNORD114.1 levels (30% vs. 57%, respectively, *p* = 0.009).

Significant associations between snoRNA levels and classical atherosclerosis risk factors are summarized in Table 1A–C.

### 2.5. SnoRNA levels and Platelet Function

To investigate whether plasma levels of snoRNAs were associated with on-treatment platelet activation, we compared in vivo and ADP-inducible expression of P-selectin and, activated GPIIb/IIIa, as well as in vivo and ADP-inducible MPA formation between patients with low vs. high snoRNA levels. Moreover, we compared residual platelet reactivity to ADP by the VASP assay (PRI, platelet reactivity index) and the total platelet count between patients with low and high snoRNA levels. High SNORD114.1 was linked to significantly lower MPA formation in vivo than low SNORD114.1 (31.18 vs. 21.95, *p* = 0.015), where a similar effect was seen in patients with high vs. low SNORD113.2 (30.29 vs. 23.47 *p* = 0.039). High SNORD113.2 and SNORD114.1 levels were associated with significantly lower MPA formation in response to ADP than low SNORD113.2 and SNORD114.1 concentrations, respectively (SNORD113.2: 54.67 vs. 45.09, *p* = 0.015; SNORD114.1: 55.22 vs. 44.65; *p* = 0.010). In contrast, high vs. low SNORD113.2 and SNORD114.1 were not associated with activated GPIIb/IIIa or P-selectin expression (Table 2).

We then used linear regression models to investigate potential correlations between SNORD113.2 and SNORD114.1 and on-treatment platelet activation. Furthermore, as both white blood cell count (WBC) and smoking were associated with snoRNAs, and may influence platelet activation, we used linear regression analyses to correct for these potential confounders. While the outcome parameters of the regression analysis (collinearity statistics, Durbin–Watson, constant variance of the residuals and normal distribution of residuals) were within an acceptable range for SNORD113.2, some parameters (Durbin–Watson and normal distribution of residuals) were sub-optimal for SNORD114.1, meaning that the *p*-values of the regression analyses for SNORD114.1 had to be interpreted with caution. Scatter plots showing all significant correlations are shown in the Appendix A.

SNORD113.2 plasma levels showed strong trends towards inverse correlations with in vivo p-selectin expression (B = −0.023, *p* = 0.055), in vivo expression of activated GPIIb/IIIa (B = −0.017, *p* = 0.080), and with the platelet count (B = −1.087, *p* = 0.079). Moreover, SNORD113.2 levels correlated inversely with in vivo and ADP-inducible MPA formation (in vivo: B = −0.441, *p* = 0.028; ADP: B = −0.653, *p* = 0.002), and with the PRI (B = −0.600, *p* = 0.029). After correction for WBC and smoking, the inverse correlations with in vivo p-selectin expression (B = −0.026, *p* = 0.033), and activated GPIIb/IIIa in vivo (B = −0.020, *p* = 0.035), both in vivo and ADP-inducible MPA formation (B = −0.524, *p* = 0.010; B = −0.709, *p* = 0.001, respectively), and the PRI (B = −0.596, *p* = 0.034) were statistically significant (Table 3).

SNORD114.1 plasma levels showed strong trends towards inverse correlations with in vivo P-selectin expression (B = −0.026, *p* = 0.066) and in vivo MPA formation (B = −0.398, *p* = 0.087), and correlated inversely with ADP-inducible MPA formation (B = −0.532, *p* = 0.037). After correction for WBC and smoking, only the inverse correlations with in vivo p-selectin expression (B−0.029, *p* = 0.048) and ADP-inducible MPA formation (B = −0.559, *p* = 0.034) were statistically significant indicating that leukocyte counts and smoking can explain some, but by far not all inverse correlations between SNORD113.2 and SNORD114.1 levels and platelet activation in PAD patients (Table 3).

When corrected for each other, inverse correlations between platelet activation and SNORD113.2, not SNORD114.1, remained. However, rather than the confounders, these two snoRNAs were most likely to be released or regulated together.

## 3. Discussion

In this study, we assessed levels of four different snoRNAs from the 14q32 noncoding RNA gene locus. We found that 14q32 snoRNAs were readily detectable in plasma samples from patients with stable PAD. However, they were not associated with the primary endpoint of our study, namely target vessel restenosis, myocardial infarction and TIA/stroke. For the most part, snoRNA levels were not consistently associated with classical risk factors for atherosclerosis either. Two of these snoRNAs however, SNORD113.2 and SNORD114.1 showed significant inverse correlations with both leukocyte counts and active smoking. More importantly, levels of SNORD113.2 and SNORD114.1 showed inverse correlations with platelet activation, which remained significant, even after correction for WBC and smoking.

In a previous study, we reported that 14q32 snoRNAs were abundantly expressed in patients with end stage PAD, but were expressed at much lower levels in healthy individuals [9]. Although all patients in the current study had intermittent claudication and no critical limb ischemia, all four snoRNAs were readily detectable, especially SNORD113.2 and SNORD114.1. 14q32 SnoRNAs therefore appear elevated in all PAD patients, not just in patients with critical limb ischemia. Although we cannot confirm this with the current study, it appears that 14q32 snoRNA levels would not be suitable as markers for disease progression in PAD.

Restenosis is a very common problem in PAD patients [17]. While the restenosis rate is generally less than 15% in patients undergoing PCI, restenosis rates can be as high as 40–50% in PAD patients undergoing angioplasty and stenting [3]. This may at least in part be attributable to higher platelet activation and a diminished response to antiplatelet therapy in PAD compared to CAD [18]. In our population, target vessel restenosis occurred in ~35% of patients. Unfortunately, 14q32 snoRNA levels were not associated with the occurrence of restenosis, nor with time to restenosis. Furthermore, snoRNAs showed no correlation with most of the classical risk factors for atherosclerosis and restenosis. We did observe correlations between SNORD112 and Lpa and between SNORD113.2 and triglyceride levels, but not with hyperlipidemia. Although statistically significant, these correlations do not appear consistent or clinically relevant.

More consistent, and likely also more relevant, associations were found between levels of SNORD113.2 and SNORD114.1 and active smoking. Active smokers had lower plasma levels of these 14q32 snoRNAs. Although smoking has been shown to decrease the risk of restenosis [7], there was no significant reduction of restenosis among smokers in our study population, even though there was a trend towards delayed time to endpoint. Whether smoking affects snoRNA expression directly or indirectly cannot be concluded from our study. However, it has been shown that nicotine affects DNA methylation, thereby noncoding RNA expression of the 14q32 locus in lung carcinoma cell lines [19]. Furthermore, previous studies together with our study reported that smoking was associated with decreased ADP-inducible platelet reactivity, i.e., a better response to clopidogrel, in patients receiving dual antiplatelet therapy following angioplasty and stenting [20,21].

SNORD113.2 and SNORD114.1 levels were also associated with WBC. This could indicate an association with systemic inflammation, however no associations were observed with CRP levels. Leukocytes play a crucial role in both the development and progression of atherosclerotic disease, including PAD. It has been shown previously that platelets can facilitate leukocyte extravasation into the atherosclerotic lesion [22]. In a recent study, we showed that a microRNA that is also transcribed from the 14q32 locus, miR-494-3p, plays an important role in platelet function and leukocyte-platelet interactions in an atherosclerosis model in mice [23]. In fact, the 14q32 locus has been implicated in platelet activation by others too [24].

With respect to platelet function, we observed significant associations between SNORD113.2 and SNORD114.1 concentrations and various parameters of platelet activation, as well as with platelet count. Higher snoRNA levels were associated with lower platelet activation and platelet counts. We adjusted our analyses for smoking and WBC. Although the association with platelet count could be attributed to WBC rather than to snoRNA levels, associations with platelet activation, and specifically with MPA formation, remained. As antiplatelet therapy is a crucial component of PAD treatment, particularly following endovascular interventions, reliable markers for platelet function and activation could be of great clinical importance.

What remains to be determined is whether platelets are indeed the main source of circulating 14q32 snoRNAs. In experiments performed for a separate study, we found that SNORD113.2 and SNORD114.1 were not expressed in platelets from healthy individuals (data not shown), but then we also found that circulating 14q32 snoRNAs were much lower in healthy individuals than in PAD patients [9]. A recent comprehensive study of snoRNA expression throughout the body indicated fibroblasts as the predominant site of 14q32 snoRNA expression [25]. We recently showed however that the vascular wall of both human arteries and veins also express 14q32 snoRNAs [10]. Future studies will have to determine whether or not 14q32 snoRNA expression is induced in platelets from patients with PAD.

A limitation of our study is that we only looked at a small selection of highly specific snoRNAs. As microarray-based platforms are not widely available for snoRNAs, this type of small RNA has been mostly ignored in studies on circulating biomarkers. However, now that RNA-sequencing is becoming more affordable and therefore increasingly being performed, there may be a stronger focus on circulating snoRNAs as biomarkers in cardiovascular disease.

In conclusion, we show here that four selected 14q32 snoRNAs were abundantly expressed in the plasma of stable PAD patients undergoing angioplasty and stenting because of intermittent claudication. Although these snoRNAs were not associated with target vessel restenosis or reocclusion in our population, they were linked to platelet activation, even after correction for smoking and WBC. These correlations may be relevant to other forms of atherosclerotic disease, such as coronary artery disease and carotid artery disease, as well.

## 4. Materials and Methods

### 4.1. Study Population

In this prospective cohort study 104 patients undergoing successful infrainguinal angioplasty with endovascular stent implantation were enrolled consecutively at the Division of Vascular Medicine of the Medical University of Vienna [6,26]. All patients had intermittent claudication classified as Rutherford stages of PAD 2–3 due to sonographically confirmed infrainguinal artery stenosis and occlusion, respectively. All patients received long-term aspirin therapy (100 mg/day), and 75 mg of clopidogrel per day for three months following angioplasty and stenting. Clinical follow-up was assessed one and two years after the percutaneous intervention. The study protocol was approved by the Ethics Committee of the Medical University of Vienna on 20 November 2007 in accordance with the Declaration of Helsinki (Project No. 126/2007) and written informed consent was obtained from all study participants.

Exclusion criteria were a known aspirin or clopidogrel intolerance (allergic reactions, gastrointestinal bleeding), a therapy with vitamin K antagonists (warfarin, phenprocoumon, acenocoumarol) or direct oral anticoagulants (dabigatran, rivaroxaban, apixaban, edoxaban), a treatment with ticlopidine, dipyridamol or nonsteroidal anti-inflammatory drugs, a family or personal history of bleeding disorders, malignant paraproteinemias, myeloproliferative disorders or heparin-induced thrombocytopenia, severe hepatic failure, known qualitative defects in thrombocyte function, a major surgical procedure one week prior to enrolment, a platelet count <100,000 or >450,000/µL and a hematocrit <30%.

The study protocol was approved by the Ethics Committee of the Medical University of Vienna in accordance with the Declaration of Helsinki and written informed consent was obtained from all study participants.

### 4.2. Blood Sampling

Blood was drawn by aseptic venipuncture from an antecubital vein using a 21-gauge butterfly needle (0.8 × 19 mm; Greiner Bio-One, Kremsmünster, Austria) one day after the percutaneous intervention, as previously described [27]. To avoid procedural deviations all blood samples were taken by the same physician applying a light tourniquet, which was immediately released, and the samples were mixed adequately by gently inverting the tubes. After the initial 3 mL of blood had been discarded to reduce procedurally induced platelet activation, blood was drawn into 3.8% sodium citrate Vacuette tubes (Greiner Bio-One; 9 parts of whole blood, 1 part of sodium citrate 0.129 M/L) for flow cytometry analyses and the VASP assay, and for RNA isolation. To avoid investigator-related variations of the results, each assay was performed by just one operator, who was blinded to clinical follow-up. Total cholesterol levels, triglycerides, LDL, HDL, Lipoprotein (a) (Lpa), CRP levels and total leukocytes were measured following standard procedures at the central laboratory of the Medical University Hospital.

### 4.3. Determination of P-Selectin Expression and Glycoprotein (GP) IIb/IIIa Activation

The expression of P-selectin and the binding of the monoclonal antibody PAC-1 to activated GPIIb/IIIa were determined in citrate-anticoagulated blood, as previously published [28,29]. In brief, whole blood was diluted in phosphate-buffered saline to obtain 20 × 10^3^/µL platelets in 20 µL, and incubated for 10 min without agonists (= in vivo P-selectin expression and activated GPIIb/IIIa), and after in vitro exposure to suboptimal concentrations of ADP (10 µL at a final concentration of 1 µM; Dynabyte, Munich, Germany). The platelet population was identified by staining with anti-CD42b (5 µL of clone HIP1, allophycocyanin labelled, final dilution 1:9; Becton Dickinson (BD), San Jose, CA, USA), and the expression of P-selectin and activated GPIIb/IIIa were determined by the binding of the monoclonal antibodies PAC-1-fluorescein (5 µL, final dilution 1:9; BD) and anti-CD62p-phycoerythrin (5 µL of clone CLB-Thromb6, final dilution 1:9; Immunotech, Beckman Coulter, Fullerton, CA, USA), respectively. Isotype-matched control antibodies were used in separate vials for the determination of non-specific binding. After 15 min of incubation in the dark, the reaction was stopped by adding 500 µL PBS and samples were acquired immediately on a FACSCalibur flow cytometer (BD) with excitation by an argon laser at 488 nm and a red diode laser at 635 nm at a rate of 200–600 events per second. FITC and PE labelled beads were used to compensate manually for the FITC signal into the PE channel and vice versa. Platelets were gated in a side scatter versus FL4 dot plot. A total of 10,000 events were acquired within this gate. Positive analysis regions for P-selectin and activated GPIIb/IIIa, respectively, were set with appropriate nonspecific controls. The gated events were further analyzed in histograms for FL-1 and FL-2 for PAC-1 and P-selectin, respectively, using the CellQuest Pro software (BD). Standard BD Calibrite beads were used for daily calibration of the cytometer.

### 4.4. Determination of Monocyte-Platelet Aggregates (MPA)

MPA were identified as previously described [28,29]. In brief, HEPES buffer (for the determination of in vivo MPA formation) or ADP (1.5 µM; for the determination of ADP-inducible MPA formation) was added to 5 µL whole blood, which had been diluted with 55 µL HEPES-buffered saline. After 15 min, monoclonal antibodies (anti-CD45-fluorescein isothiocyanate (BD), anti-CD41-peridinin chlorophyll protein, (clone P2, Immunotech, Marseilles, France), and anti-CD14-allophycocyanin), or istoype-matched controls were added. After 20 min, samples were diluted with FACSlysing solution and at least 5000 CD45+ events were acquired immediately. Monocytes were identified as CD14+ and the percentage of CD14+CD41+ events was recorded as MPA.

### 4.5. Vasodilator-Stimulated Phosphoprotein (VASP) Phosphorylation Assay

For determination of the platelet reactivity index (PRI), the extent of VASP phosphorylation was measured by geometric mean fluorescence intensity (MFI) values in the presence of PGE1 without (T1) or with ADP (T2) as previously described [30]. After subtraction of the background fluorescence from the corresponding fluorescence values, the PRI (%) was calculated according to the following formula:PRI % = [T1 (PGE1) − T2 (PGE1+ADP)/T1 (PGE1)] × 100(1)

### 4.6. Clinical Endpoints

Clinical follow-up was assessed at regular visits of the study participants to the outpatient department of the Division of Vascular Medicine at the Medical University of Vienna and via telephone calls, respectively. The primary endpoint was defined as the composite of the first occurrence of any of the following events: nonfatal myocardial infarction (MI), nonfatal stroke or transient ischemic attack (TIA), cardiovascular death, and sonographically confirmed >80% target vessel restenosis or reocclusion within two years after peripheral angioplasty and stenting [26].

### 4.7. RNA Isolation and SnoRNA Measurements

Total RNA was isolated from 200 µL citrate-anticoagulated plasma, using the Maxwell^®^ RSC simplyRNA Tissue Kit and the Maxwell^®^ RSC instrument (Promega GmbH, Mannheim, Germany), according to the manufacturer’s protocol. A DNase treatment was included to prevent DNA contamination of the samples. RNA concentration and purity were checked by Nanodrop (ThermoFisher Scientific, Vienna, Austria). RNA was reverse transcribed using the high-capacity RNA to cDNA kit (ThermoFisher Scientific). We then measured the plasma levels of four human snoRNAs; SNORD112, SNORD113.2, SNORD113.6 and SNORD114.1 by quantitative PCR on the Light cycler 480 Real-time PCR System (Roche Diagnostics GmbH), using the GoTaq qPCR kit (Promega). SnoRNA levels were expressed as expression relative to snU6. Primer sequenced were as published previously [9,10,31].

### 4.8. Statistical Analyses

All analyses were preformed using SPSS 26 software. Associations between categorical variables, including primary clinical endpoints and smoking, and snoRNA levels were determined using Mann-Whitney U tests or ANOVA, comparing snoRNAs levels between groups. Patients were also distributed into two groups for each snoRNA, having either below median (Low) or above median (High) expression. Associations between low or high expression and categorical variables were analysed using the Crosstabs function and Pearson Chi-Square test was used to determine statistical significance. Finally, Kaplan–Meier survival plots were generated to investigate potential associations between snoRNA levels and time to clinical endpoint.

Associations between snoRNAs and continuous variables were determined using linear regression. Linear regression was also used to correct for potential confounding factors. We also compared continuous variables between individuals with either low or high snoRNA levels using Mann-Whitney U tests.

## Figures and Tables

**Figure 1 ijms-20-05975-f001:**
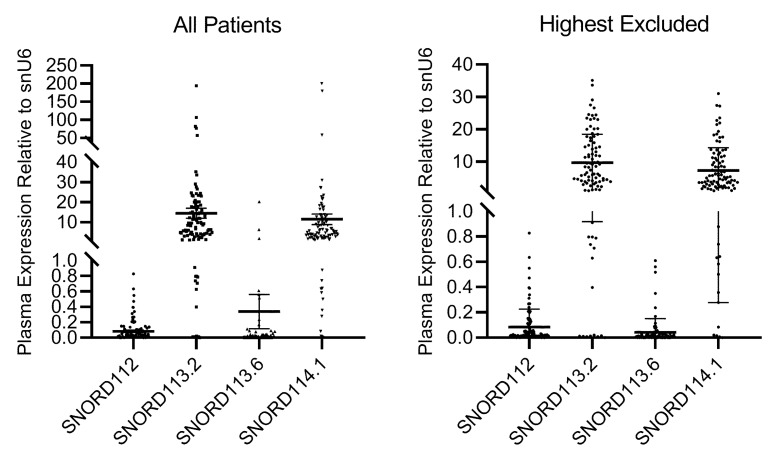
Distribution of snoRNA expression relative to snU6 in (**left panel**) all patients and in (**right panel**) the subset of patients included in all further analyses.

**Table ijms-20-05975-t001a:** (**A**)

**snoRNA**	**Hypertension**	***n***	**Mean**	**∆**	***p***
SNORD112	Yes	8	0.07	0.01	0.99
	No	93	0.08		
SNORD113.2	Yes	7	9.37	0.36	0.90
	No	87	9.72		
SNORD113.6	Yes	7	0.08	0.04	0.34
	No	85	0.04		
SNORD114.1	Yes	8	9.59	2.48	0.37
	No	90	7.11		
	**Hyperlipidemia**	***n***	**Mean**	**∆**	***p***
SNORD112	Yes	8	0.07	0.01	0.39
	No	93	0.08		
SNORD113.2	Yes	7	9.37	0.36	0.53
	No	87	9.72		
SNORD113.6	Yes	7	0.08	0.04	0.43
	No	85	0.04		
SNORD114.1	Yes	8	9.59	2.48	0.51
	No	90	7.11		
	**Diabetes Mellitus**	***n***	**Mean**	**∆**	***p***
SNORD112	Yes	65	0.074	0.03	0.21
	No	36	0.10		
SNORD113.2	Yes	61	9.15	1.56	0.24
	No	33	10.71		
SNORD113.6	Yes	58	0.03	0.03	0.90
	No	34	0.06		
SNORD114.1	Yes	65	6.97	1.01	0.94
	No	33	7.98		
	**Smoking**	***n***	**Mean**	**∆**	***p***
SNORD112	Yes	57	0.08	0.01	0.41
	No	44	0.09		
SNORD113.2	Yes	51	10.32	1.35	0.35
	No	43	8.97		
SNORD113.6	Yes	52	0.05	0.02	0.80
	No	40	0.03		
SNORD114.1	Yes	56	8.68	3.2	0.031
	No	42	5.48		

**Table ijms-20-05975-t001b:** (**B**)

snoRNA	Risk Factor	B	*p*
SNORD112	Lpa	−0.228	0.025
SNORD113.2	Triglycerides	0.219	0.034
SNORD113.2	WBC	−0.746	0.098
SNORD113.6	WBC	0.012	0.026

**Table ijms-20-05975-t001c:** (**C**)

**Risk Factor**	**SNORD112**	***n***	**Mean**	**∆**	***p***
Lpa	Low		72.47	21.67	0.038
	High		50.80		
	**SNORD113.2**	***n***	**Mean**	**∆**	***p***
Triglycerides	Low		155.49	58.07	0.27
	High		213.56		
WBC	Low		9.19	0.70	0.10
	High		8.49		
	**SNORD114.1**	***n***	**Mean**	**∆**	***p***
WBC	Low		9.12	0.73	0.057
	High		8.39		

**Table 2 ijms-20-05975-t002:** SnoRNA levels and platelet activation.

**Parameter of Platelet Function**	**SNORD113.2**	***n***	**Mean**	**∆**	***p***
In vivo P-Selectin	Low	46	3.60	0.31	0.07
	High	49	3.29		
ADP-inducible P-selectin	Low	46	15.23	0.27	0.50
	High	49	14.96		
In vivo activated GPIIb/IIIa	Low	46	2.99	0.21	0.18
	High	49	2.78		
ADP-inducible GPIIb/IIIa	Low	46	12.02	0.25	0.69
	High	49	11.77		
In vivo MPA	Low	38	30.29	6.82	0.039
	High	42	23.47		
ADP-inducible MPA	Low	38	54.67	9.58	0.015
	High	42	45.09		
PRI	Low	48	49.34	7.54	0.09
	High	50	41.80		
**Parameter of Platelet Function**	**SNORD114.1**	***n***	**Mean**	**∆**	***p***
In vivo P-Selectin	Low	49	3.60	0.31	0.11
	High	48	3.29		
ADP-inducible P-selectin	Low	49	16.65	2.46	0.17
	High	48	14.20		
In vivo activated GPIIb/IIIa	Low	49	2.92	0.05	0.51
	High	48	2.87		
ADP-inducible GPIIb/IIIa	Low	49	12.13	0.03	0.90
	High	48	12.10		
In vivo MPA	Low	44	31.18	9.23	0.015
	High	38	21.95		
ADP-inducible MPA	Low	44	55.22	10.57	0.010
	High	39	44.65		
PRI	Low	50	48.79	5.80	0.24
	High	50	42.99		

**Table 3 ijms-20-05975-t003:** SnoRNA levels and platelet activation; linear regression.

**SNORD113.2**				
**Parameter of Platelet Function**	**B**	***p***	**B_Adjusted_**	***p*_Adjusted_**
In vivo P-Selectin	−0.023	0.055	−0.026	0.033
ADP-inducible P-selectin	−0.078	0.557	−0.112	0.406
In vivo GPIIb/IIIa	−0.017	0.080	−0.020	0.035
ADP-inducible GPIIb/IIIa	−0.062	0.499	−0.098	0.283
In vivo MPA	−0.441	0.028	−0.524	0.010
ADP-inducible MPA	−0.653	0.002	−0.709	0.001
PRI	−0.600	0.029	−0.596	0.034
**SNORD114.1**				
**Parameter of Platelet Function**	**B**	***p***	**B_Adjusted_ ***	***p*_Adjusted_ ***
In vivo P-Selectin	−0.026	0.066	−0.029	0.048
ADP-inducible P-selectin	−0.141	0.393	−0.192	0.255
In vivo GPIIb/IIIa	−0.006	0.613	−0.012	0.296
ADP-inducible GPIIb/IIIa	−0.029	0.796	−0.071	0.526
In vivo MPA	−0.398	0.087	−0.352	0.137
ADP-inducible MPA	−0.532	0.037	−0.559	0.034
PRI	−0.480	0.161	−0.515	0.147

* Adjusted for smoking and WBC.

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
