# Peer review of "Plasma Levels of snoRNAs are Associated with Platelet Activation in Patients with Peripheral Artery Disease"

_ijms, 2019, doi:10.3390/ijms20235975_

Round 1

Reviewer 1 Report

This manuscript examines the expression levels of 4 SNORD RNAs in the plasma of individuals suffering from PAD and suggests that expression of two of the four SNORDs is linked to platelet activation.

The manuscript is well written but I did find the results a little hard going with all the numbers that are discussed one after the other.  Having said that, I am not really sure what to suggest in terms of how to make it a little more user friendly.

You mention in your discussion that a study has looked at SNORD expression in the platelets of healthy individuals.  Given your results,would it be possible to look at expression levels in individuals suffering PAD which may then help support a possible involvement in platelet activation.

Author Response

This manuscript examines the expression levels of 4 SNORD RNAs in the plasma of individuals suffering from PAD and suggests that expression of two of the four SNORDs is linked to platelet activation.

1. The manuscript is well written but I did find the results a little hard going with all the numbers that are discussed one after the other.  Having said that, I am not really sure what to suggest in terms of how to make it a little more user friendly.

We realize that there are a lot of data presented as dry numbers in the results section. However, these are the data and not to include them in the results section would not do the study full justice. However, also in response to reviewer 2, we have now added the regression graphs as supplemental data in the revised manuscript. The graphs will facilitate the interpretation of the presented data.

2. You mention in your discussion that a study has looked at SNORD expression in the platelets of healthy individuals.  Given your results, would it be possible to look at expression levels in individuals suffering PAD which may then help support a possible involvement in platelet activation.

We appreciate the suggestion to measure snoRNA levels in platelets of patients with peripheral artery disease. Unfortunately, we do not have such material at our disposal at this time. We have now commented on this opportunity for future studies in the discussion section of the revised manuscript (page 8, lines 250-251).

Reviewer 2 Report

Generally speaking, the data analysis in this manuscript supports the conclusion. These selected snoRNA although not strongly correlated with most PAD related factors, but showed certain level of association with the disease condition. There are some questions needs address and revise before further consideration for publication.

The authors clearly are experts in PAD and related cardiovascular disease clinical treatments and studies. However, it might still be not completely correct stating revascularization interventions (stent or by pass) are the sole treatment opinions for PAD. Currently, according to AHA guideline and many insurance policy, treadmill supervised exercise is the first line before these invasive treatments. Although, on the other hand, there are some recent strong evidence from Drs. Pipinos/Casale/Li group suggesting current first line exercise therapy needs modification, which were presented in both AHA Vascular Discovery and SVS annual meeting as poster and oral speeches. Before any statistical test, authors should better provide information about the characters of the selected data. Are all of these data normally distributed and all meets the assumptions for the used statistical methods? If not, some of the result should be analyzed with different methods. Even yes to all of these data, this statement is needed. Please provide the rationale using median as the separation point between low and high snoRNA values. Median is not necessarily the cut off point diving data into subgroups. Maybe cluster analysis is a better opinion. Some of the significance could be compromise by using simply median. For linear correlation, firstly, are these data parametric to use this analysis? Secondly, for the significant linear regressions, the plot figure should be helpful in additional than just R and P values. Lastly, these snoRNA could be simply atherosclerosis related markers instead of PAD specific ones. Current data might not be strong enough to claim this general conclusion, unless compared with other cardiovascular diseases including other atherosclerosis patients serum values. 

Author Response

Generally speaking, the data analysis in this manuscript supports the conclusion. These selected snoRNA although not strongly correlated with most PAD related factors, but showed certain level of association with the disease condition. There are some questions needs address and revise before further consideration for publication.

1. The authors clearly are experts in PAD and related cardiovascular disease clinical treatments and studies. However, it might still be not completely correct stating revascularization interventions (stent or by pass) are the sole treatment opinions for PAD. Currently, according to AHA guideline and many insurance policy, treadmill supervised exercise is the first line before these invasive treatments. Although, on the other hand, there are some recent strong evidence from Drs. Pipinos/Casale/Li group suggesting current first line exercise therapy needs modification, which were presented in both AHA Vascular Discovery and SVS annual meeting as poster and oral speeches.

The reviewer is correct to state that supervised walking therapy is probably the most important and most efficient form of therapy for PAD patients. We have now mentioned supervised walking therapy in the introduction section of the revised manuscript (page 2, lines 43-44).

2. Before any statistical test, authors should better provide information about the characters of the selected data. Are all of these data normally distributed and all meets the assumptions for the used statistical methods? If not, some of the result should be analyzed with different methods. Even yes to all of these data, this statement is needed.

We thank the reviewer for this valuable suggestion. We performed Shapiro-Wilk tests and concluded that not all data follow a normal distribution. We therefore replaced all t-tests in the manuscript by Mann-Whitney tests. Although some p-values (exact 2-tailed) changed slightly, and a trend for SNORD113.2 (with in vivo monocyte-platelet aggregation) became significant, there were no further changes in the outcomes.

3. Please provide the rationale using median as the separation point between low and high snoRNA values. Median is not necessarily the cut-off point dividing data into subgroups. Maybe cluster analysis is a better option. Some of the significance could be compromise by using simply median.

There are many ways to dichotomize groups. In clinical practice a cut-off value is often appreciated. We chose to use the median as, other than the mean, this value is not impacted by a non-normal distribution of the data. We also looked at quartiles, and found similar outcomes, but for the sake of readability, we chose to only show the median cut-off data.

4. For linear correlation, firstly, are these data parametric to use this analysis? Secondly, for the significant linear regressions, the plot figure should be helpful in additional than just R and P values.

We checked the suitability of our data for linear regression. We checked for multicollinearity of the predictors, and found no problems. We also performed a Durbin-Watson test to check for independency of the residuals. While there were no problems for SNORD113.2, for SNORD114.1, the Durbin-Watson statistic was just below 1 for all regression analyses, which is rather low. Also the distribution of the residuals was not completely normal. This means that although the regression analyses are reliable, the p-values for SNORD114.1 should be interpreted with caution. We have added this information to the results section of the revised manuscript (page 6; lines 161-165). As the associations were generally stronger for SNORD113.2 than for SNORD114.1, this does not change the conclusions of our study.

We have now added the scatter-plots for all significant correlations between SNORD113.2 and SNORD114.1 with platelet parameters as Supplemental Data file.

5. Lastly, these snoRNA could be simply atherosclerosis related markers instead of PAD specific ones. Current data might not be strong enough to claim this general conclusion, unless compared with other cardiovascular diseases including other atherosclerosis patients serum values. 

This is of course a very valid point. As we looked in PAD patients only and not in patients with other forms of atherosclerotic disease, we were cautious not to draw our conclusions beyond PAD. However, we have now added the following sentence to our conclusions (page 9; lines 265-266):

“These correlations may be relevant in other forms of atherosclerotic disease, such as coronary artery disease and carotid artery disease, as well.”
